# GRADIENT-OPTIMIZED CONTRASTIVE LEARNING

## ABSTRACT

Contrastive learning is a crucial technique in representation learning, producing robust embeddings by distinguishing between similar and dissimilar pairs. In this paper, we introduce a novel framework, *Gradient-Optimized Contrastive Learning (GOAL)*, which enhances network training by optimizing gradient updates during backpropagation as a bilevel optimization problem. Our approach offers three key insights that set it apart from existing methods: (1) Contrastive learning can be seen as an approximation of a one-class support vector machine (OC-SVM) using multiple neural tangent kernels (NTKs) in the network's parameter space; (2) Hard triplet samples are vital for defining support vectors and outliers in OC-SVMs within NTK spaces, with their difficulty measured using Lagrangian multipliers; (3) Contrastive losses like InfoNCE provide efficient yet dense approximations of sparse Lagrangian multipliers by implicitly leveraging gradients. To address the computational complexity of GOAL, we propose a novel contrastive loss function, *Sparse InfoNCE (SINCE)*, which improves the Lagrangian multiplier approximation by incorporating hard triplet sampling into InfoNCE. Our experimental results demonstrate the effectiveness and efficiency of SINCE in tasks such as image classification and point cloud completion. **Demo code is attached in the supplementary file.**

## 1 INTRODUCTION

Contrastive learning (Chopra et al., 2005; Hadsell et al., 2006) has become one of the dominant methods in representation learning. Typically, contrastive learning constructs positive pairs and negative pairs by creating two augmented views of the same image. The goal is to bring the embeddings of positive pairs closer and push those of negative pairs apart in the latent space, often optimized using a loss function such as InfoNCE (Van den Oord et al., 2018; Chen et al., 2020a).

**Motivation.** To better understand contrastive learning, we start by analyzing the impacts of positive and negative samples on the gradients during backpropagation in training. We discover that recent contrastive losses often result in bounded positive weights for linear combinations of triplet gradient features in stochastic gradient descent (SGD). For instance, Tian (2022) recently proposed a family of $(\phi, \psi)$-contrastive losses defined as $\ell_{\phi,\psi} = \sum_x \phi \left( \sum_{x^-} \psi \left( f(x, x^+, x^-; \omega) \right) \right)$, where the scalar functions $\phi$ and $\psi$ are increasing monotonically and differentiable. The function $f(x, x^+, x^-; \omega) = \frac{1}{2}[\|h(x; \omega) - h(x^+; \omega)\|^2 - \|h(x; \omega) - h(x^-; \omega)\|^2]$ measures the distance difference between the positive and negative pairs. We list some examples in Table 1 where $\alpha_{x^-}$ denotes the weights for feature combination during learning. As we see, all the $\alpha_{x^-}$'s are positive and the summation over negative samples for each loss is no greater than one.

This behavior raises concerns about the effectiveness and robustness of the gradients in contrastive learning because useful (hard) negative samples can be easily buried among many non-useful (easy) negative samples, leading to similar weights for generating gradients. Such concerns have recently garnered increased attention. For instance, Wang & Liu (2021) claimed that "A well-designed contrastive loss should have some extent of tolerance to the closeness of semantically similar samples," and thus proposed an explicitly hard negative sampling method by *filtering out uninformative* negative samples. Chuang et al. (2020) proposed a *debiased* contrastive learning method that corrects for the sampling of same-label datapoints by thresholding in the contrastive loss. Motivated by these works, in this paper we aim to address the following question:

*How should we optimize the gradients in contrastive learning, effectively and efficiently?*

Table 1: Some examples of $(\phi, \psi)$-contrastive losses with corresponding analytical expressions.

| Contrastive Loss | $\phi(x)$ | $\psi(x)$ | $\alpha_{x^-}$: gradient feature weights |
|---|---|---|---|
| InfoNCE (Van den Oord et al., 2018) | $\tau \log(\epsilon + x)$ | $\exp\{\frac{x}{\tau}\}$ | $\dfrac{\exp\{\frac{1}{\tau} f(x,x^+,x^-;\omega)\}}{\epsilon+\sum_{x^-} \exp\{\frac{1}{\tau} f(x,x^+,x^-;\omega)\}}$ |
| MINE (Belghazi et al., 2018) | $\log(x)$ | $\exp\{x\}$ | $\dfrac{\exp\{f(x,x^+,x^-;\omega)\}}{\sum_{x^-} \exp\{f(x,x^+,x^-;\omega)\}}$ |
| Soft Triplet (Tian et al., 2020c) | $\tau \log(1 + x)$ | $\exp\{\frac{x}{\tau} + \epsilon\}$ | $\dfrac{\exp\{\frac{1}{\tau} f(x,x^+,x^-;\omega)\}}{\exp\{-\epsilon\}+\sum_{x^-} \exp\{\frac{1}{\tau} f(x,x^+,x^-;\omega)\}}$ |
| $N + 1$ Tuplet (Sohn, 2016) | $\log(1 + x)$ | $\exp\{x\}$ | $\dfrac{\exp\{f(x,x^+,x^-;\omega)\}}{1+\sum_{x^-} \exp\{f(x,x^+,x^-;\omega)\}}$ |

**Approach.** In contrast to the literature, we propose a novel framework, namely *Gradient-Optimized Contrastive Learning (GOAL)*, to learn to optimize gradients in backpropagation. Specifically, we formulate the lower-level optimization problem as a one-class support vector machine (OC-SVM) (Schölkopf et al., 1999) in a neural tangent kernel (NTK) (Jacot et al., 2018) space to determine the weights (*i.e.,* Lagrangian multipliers) for the upper-level summation loss over the triplets. We hypothesize that these weights may be taken as sub-optimal solutions to the dual of these kernel machines that explicitly learn to maximize the triplet separation in each NTK space. This interpretation is motivated by the strong connections between the dual form of OC-SVM and the linear combination weights for the gradients (*e.g.,* $\alpha_{x^-}$ in Table 1) in contrastive learning. Our analysis also implies that truly hard negative samples (in the context of triplets, rather than pairs as in traditional methods) should be defined as the support vectors and outliers of OC-SVMs in the NTK spaces, rather than in the spatial domain of images or the output space of the network. To address the computational issue in GOAL due to the nature of bilevel optimization for large-scale learning, we further propose a new contrastive loss, namely, *Sparse InfoNCE (SINCE)*, for better approximations of Lagrangian multipliers based on InfoNCE with hard triplet sampling. We demonstrate its effectiveness and efficiency in the tasks of image classification and point cloud completion, with significant improvements.

**Contributions.** In summary, our key contributions are as follows:

- We propose a new contrastive learning framework, GOAL, based on bilevel optimization that learns to optimize gradients in backpropagation for training networks. Our approach provides novel insights to understand contrastive learning from a perspective of sparse kernel machines.
- We propose a new contrastive loss, SINCE, to mitigate the computational issue in bilevel optimization by approximating the Lagrangian multipliers using InfoNCE with hard triplet sampling.
- We demonstrate superior performance in both image classification and point cloud completion, showcasing the effectiveness and efficiency of our approach.

## 2 RELATED WORK

**Contrastive Learning.** Learning representations from unlabeled data in a contrastive way has been one of the most competitive research fields (Van den Oord et al., 2018; Hjelm et al., 2018; Wu et al., 2018; Tian et al., 2020a; Sohn, 2016; Chen et al., 2020a; Jaiswal et al., 2020; Li et al., 2020b; He et al., 2020; Chen et al., 2020c;b; Bachman et al., 2019; Misra & Maaten, 2020; Caron et al., 2020) where contrastive loss optimizes data representations by aligning the two views of the same image (*i.e.,* positive pairs) while pushing different images (*i.e.,* negative pairs) away. A large number of works in contrastive learning are about how to augment the data. Empirically, positive pairs could be different modalities of a signal (Arandjelovic & Zisserman, 2018; Tian et al., 2020a; Tschannen et al., 2020) or different augmented samples of the same image *e.g.,* color distortion and random crop (Chen et al., 2020a;c; Grill et al., 2020). Tian et al. (2020b) suggested generating positive pairs with the "InfoMin principle" so that the generated positive pairs maintain the minimal information necessary for downstream tasks. Selvaraju et al. (2021); Peng et al. (2022); Mishra et al. (2021); Li et al. (2022) proposed selecting meaningful but not fully overlapped contrastive crops with guidance such as attention maps or object-scene relations. Shen et al. (2020) empirically demonstrated that introducing extra convex combinations of data as positive augmentation improves representation learning. Similar mixing data strategies could be found in (Lee et al., 2020; Kim et al., 2020; Verma et al., 2021; Li et al., 2020a; Ren et al., 2022). In addition to exploring positive augmentation, some recent work

also focuses on negative data selection in contrastive learning. Typically, negative samples are drawn uniformly from the training data. Based on the argument that not all negatives are true negatives, Chuang et al. (2020); Robinson et al. (2020) developed debiased contrastive losses to assign higher weights to "harder" negative samples. Wang & Liu (2021) proposed an explicit way to select hard negative samples that are similar to the positives. To provide more meaningful negative samples, Kalantidis et al. (2020) studied the Mixup (Zhang et al., 2017) strategy in latent space to generate hard negatives. Hu et al. (2021) proposed learning a set of negative adversaries directly. Ge et al. (2021) generated negative samples by texture synthesis or selecting non-semantic patches from existing images. Yue et al. (2024) studied hard negative samples in the hyperbolic space and proposed a new contrastive loss by considering both Euclidean and hyperbolic spaces.

**Sparse Kernel Machines.** A sparse kernel machine is a type of statistical learning algorithm that focuses on using a subset of training data to make predictions. This approach is beneficial in scenarios where the dataset is large, as it helps reduce computational complexity and improve efficiency. OC-SVMs (Schölkopf et al., 1999; Tax & Duin, 1999; Sain, 1996; Schölkopf et al., 2001; Tax & Duin, 2004; Tax, 2002), a classical one-class learning algorithm, are frequently used in outlier or novelty detection (Pimentel et al., 2014; Chandola, 2007; Ratsch et al., 2002) to detect if a test sample belongs to the same distribution of training data. For instance, Tax & Duin (1999) proposed minimizing the volume of a hypersphere that contains as many as possible of the "normal" training data, which has been shown to be equivalent to (Schölkopf et al., 2001) for certain kernels. Some good surveys are provided in (Subrahmanya & Shin, 2009; Li et al., 2020c). Particularly, max-margin based contrastive learning (Chen et al., 2021; Shah et al., 2022) have been studied as well.

**Point Cloud Completion.** In computer vision, this refers to an important and challenging task of inferring the complete 3D shape of an object or scene from incomplete raw 3D point clouds. Recently, many deep learning approaches have been developed for this task. For instance, PCN (Yuan et al., 2018), the first deep neural network for point cloud completion, extracts global features directly from point clouds and then generates points using the folding operations from FoldingNet (Yang et al., 2018). Zhang et al. (2020) proposed extracting multiscale features from different network layers to capture local structures and improve performance. Attention mechanisms such as Transformer (Vaswani et al., 2017) excel at capturing long-term interactions. Accordingly, SnowflakeNet (Xiang et al., 2021), PointTr (Yu et al., 2021), and SeedFormer (Zhou et al., 2022) accentuate the decoder component by incorporating Transformer designs. PointAttN (Wang et al., 2022) is conceived entirely on Transformer foundations. In particular, Lin et al. (2023) proposed an InfoCD loss by introducing contrastive learning into point cloud completion, achieving the state-of-the-art performance.

# 3 GOAL: GRADIENT-OPTIMIZED CONTRASTIVE LEARNING

## 3.1 PRELIMINARY

**Learning with InfoNCE.** We denote $x \in \mathcal{X}, x^+ \in \mathcal{X}^+, x^- \in \mathcal{X}^-$ as an archor sample and its positive and negative samples, respectively. We further denote $h(x; \omega) : \mathcal{X} \times \Omega \to \mathbb{R}^d$ as a differentiable function that is implemented by a neural network and parametrized by $\omega \in \Omega$, and

$$f_{\tau,\tau'}(x, x^+, x^-; \omega) = \frac{1}{\tau} d(x^+, x; \omega) - \frac{1}{\tau'} d(x^-, x; \omega) \qquad (1)$$

as a distance measure for the triplet $(x, x^+, x^-)$ with some form of pairwise distance measure $d$, where $\tau, \tau' \geq 0$ denote two predefined scalars. Note that the smaller $f_{\tau,\tau'}(x, x^+, x^-; \omega)$ is, the better the separation between the positive and negative pairs. By defining $d(\cdot, x; \omega) = \|h(\cdot; \omega) - h(x; \omega)\|_2^2$ in Equation (1), the InfoNCE loss in (Van den Oord et al., 2018) can be written as follows:

$$\ell(\omega) = \mathbb{E}_x\Big[\ell_\tau(x; \omega)\Big] = \mathbb{E}_x\left[\log \sum_{x^-} \exp\left\{f_{\tau,\tau'}(x, x^+, x^-; \omega)\right\}\right], \qquad (2)$$

where only one positive sample is considered and $\mathbb{E}$ denotes the expectation operator. Now based on this equation, we can compute the gradients in backpropagation during training as

$$\nabla \ell_\tau(x; \omega) = \sum_{x^-} \alpha_{x^-} \nabla f_{\tau,\tau}(x, x^+, x^-; \omega), \text{ where } \alpha_{x^-} = \frac{\exp\{f_{\tau,\tau}(x, x^+, x^-; \omega)\}}{\sum_{x^-} \exp\{f_{\tau,\tau}(x, x^+, x^-; \omega)\}}. \qquad (3)$$

Clearly, it holds that $0 \leq \alpha_{x^-} \leq 1, \sum_{x^-} \alpha_{x^-} = 1$. Therefore, $\nabla \ell_\tau(x; \omega)$ computes the mean of the gradients $\nabla f_{\tau,\tau}(x, x^+, x^-; \omega)$ from all positive and negative samples *w.r.t.* $x$, and $\nabla \ell_\tau(\omega) = \mathbb{E}_x[\nabla \ell_\tau(x; \omega)]$ computes the mean of $\nabla \ell_\tau(x; \omega)$ over $x$. All the expressions of $\alpha_{x^-}$'s in Table 1 are computed in a similar way given different objectives.

### 3.2 OUR BILEVEL MODEL

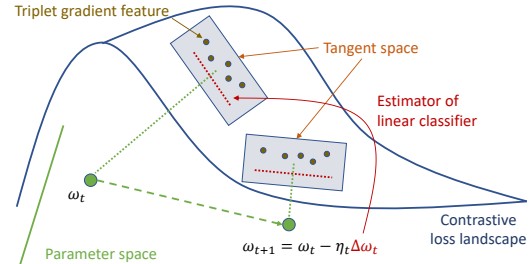

In Figure 1, we illustrate a geometric view of SGD based on a *local linear approximation* of the loss landscape at each parameter update. The loss landscape is parameterized by the network parameter $\omega$, and at each update $\omega_t$, a neural tangent space is constructed by taking triplets $\{(x, x^+, x^-)\}$ as input to generate *triplet gradient features* $\nabla f_{\tau,\tau'}(x, x^+, x^-; \omega_t)$, and then the gradient $\Delta\omega_t$ is computed by a linear combination of such triplet features, *i.e.*, $\Delta\omega_t = \sum_{(x,x^+,x^-)} \alpha^{(t)}_{(x,x^+,x^-)} \nabla f_{\tau,\tau'}(x, x^+, x^-; \omega_t)$, where $\alpha^{(t)}_{(x,x^+,x^-)}$ stands for a sample weight at the $t$-th iteration in SGD.

Figure 1: Illustration of local linear approximation of a contrastive loss landscape during training with SGD. The gradient $\Delta\omega$ is often a linear combination of triplet gradient features in the tangent space, and we show that such increments may be interpreted as approximations of linear OC-SVMs.

**Motivation: Sample weights for gradients and network weights may be fully coupled.** When the calculation of each $\alpha^{(t)}_{(x,x^+,x^-)}$ relies on the triplet features $\nabla f_{\tau,\tau'}(x, x^+, x^-; \omega_t)$, it becomes evident that $\alpha^{(t)}_{(x,x^+,x^-)}$ is a function of $\omega_t$. Consequently, training can be iteratively performed by optimizing $\omega$ towards a specific objective. Indeed, all the contrastive losses in Table 1 are designed in such a way that each $\alpha_{(x,x^+,x^-)}$ explicitly depends on $\omega$, as shown in Equation (3). Now the question is:

*What if $\alpha_{(x,x^+,x^-)}$ does not have an explicit form of $\omega$?*

To answer this question, we propose using bilevel optimization (Colson et al., 2007), where one problem is embedded (nested) within another, to model the dependency between the sample weights for gradients and network weights. In this structure, the *upper-level (UL)* problem is influenced by the *optimal* parameters from the *lower-level (LL)* problem, whereas the LL problem is influenced by the *non-optimal* parameters from the UL problem. In our model, we use the UL problem to update network weights, and the LL problem to learn optimal gradients for SGD.

**Upper-level Objective.** At the early age of contrastive learning, the losses such as (Chopra et al., 2005; Schroff et al., 2015) always favor sparse samples for learning. For instance, the triplet loss (Schroff et al., 2015) is defined as $\ell_{triplet}(x, x^+, x^-; \omega) = \max\left\{0, f_{1,1}(x, x^+, x^-; \omega) + \epsilon\right\}$, where $\epsilon \geq 0$ is a predefined parameter to control the minimum offset between distances of similar and dissimilar pairs. In fact, triplet loss is a variant of the hinge loss commonly used in SVMs. Regarding gradient calculation, the triplet loss assigns a combination weight of either 0 or 1 to the gradient of each triplet, which differs from modern contrastive losses such as InfoNCE. Considering these, we propose the following UL objective that involves the optimal solution $\{\alpha^*_{ijk}\}$ from the LL problem to model the sample weights for gradients explicitly:

$$\min_\omega \sum_{i,j,k} \alpha^*_{ijk} f_{\tau,\tau'}(x_i, x^+_{ij}, x^-_{ik}; \omega), \qquad (4)$$

where $i, j, k$ denote the $i$-th anchor, its $j$-th positive and $k$-th negative samples, respectively. In this way, we can control the gradients based on these sample weights in SGD.

**Lower-level Objective.** Recall that at the $t$-th iteration in SGD, the gradient $\Delta\omega_t$ can be represented as a linear combination of triplet gradient features $\nabla f_{\tau,\tau'}(x, x^+, x^-; \omega_t)$ with weights $\alpha^{(t)}_{(x,x^+,x^-)}$. This reminds us of the classic representer theorem (Dinuzzo & Schölkopf, 2012) for kernel methods, and motivates us to learn $\Delta\omega_t$ based on local linear approximation, namely, $f_{\tau,\tau'}(x, x^+, x^-; \omega_t - \Delta\omega_t) \approx f_{\tau,\tau'}(x, x^+, x^-; \omega_t) - \Delta\omega_t^T \nabla f_{\tau,\tau'}(x, x^+, x^-; \omega_t)$ where $(\cdot)^T$ denotes the matrix transpose operator. We expect that after the update, the value of $f_{\tau,\tau'}(x, x^+, x^-; \omega_t - \Delta\omega_t)$ could be no bigger than a

threshold $\rho_t$. Motivated by one-class support vector machine (OC-SVM) in (Schölkopf et al., 1999), we propose the following regularized OC-SVM as our LL objective:

$$\min_{\Delta\omega_t, \rho_t, \{\xi_{ijk}^{(t)}\}} \frac{1}{2}\|\Delta\omega_t\|^2 + \rho_t + C\sum_{i,j,k}\xi_{ijk}^{(t)}, \tag{5}$$

$$\text{s.t. } f_{\tau,\tau'}(x_i, x_{ij}^+, x_{ik}^-; \omega_t) - \Delta\omega_t^T\nabla f_{\tau,\tau'}(x_i, x_{ij}^+, x_{ik}^-; \omega_t) \le \rho_t + \xi_{ijk}^{(t)}, \xi_{ijk}^{(t)} \ge 0, \forall i, \forall j, \forall k, \forall t,$$

with a predefined constant $C \ge 0$ and a set of slack variables $\{\xi_{ijk}^{(t)}\}$.

**Bilevel Formulation.** As we discussed before, the sample weights for gradients, $\alpha$, and the network weights, $\omega$, are coupled, and one can be optimized alternatively by fixing the other (a widely used technique for solving bilevel optimization (Xiao et al., 2024)). Therefore, by incorporating our UL objective in Equation (4) and the dual form of our LL objective in Equation (5), we propose the following bilevel optimization problem for contrastive learning:

$$\omega^* \in \arg\min_{\omega} \sum_{i,j,k} \alpha_{ijk}^* f_{\tau,\tau'}(x_i, x_{ij}^+, x_{ik}^-; \omega), \tag{6}$$

$$\text{s.t. } \{\alpha_{ijk}^*\} \in \arg\min_{\{\alpha_{ijk}\}} \left\{ \frac{1}{2}\sum_{jk,j'k'} \alpha_{ijk}\kappa_{\omega^*}(\mathcal{X}_{ijk}, \mathcal{X}_{ij'k'})\alpha_{ij'k'} - \sum_{j,k}\alpha_{ijk}f_{\tau,\tau'}(x_i, x_{ij}^+, x_{ik}^-; \omega^*) \right\}$$

$$\text{s.t. } \sum_{j,k}\alpha_{ijk} = 1, 0 \le \alpha_{ijk} \le C, \forall i, \forall j, \forall k,$$

where for simplicity, $\mathcal{X}_{ijk} = \{x_i, x_{ij}^+, x_{ik}^-\}, \mathcal{X}_{ij'k'} = \{x_i, x_{ij'}^+, x_{ik'}^-\}$ stand for two triplets, respectively, and $\kappa_{\omega^*}(\mathcal{X}_{ijk}, \mathcal{X}_{ij'k'}) = \nabla f_{\tau,\tau'}(x_i, x_{ij}^+, x_{ik}^-; \omega^*)^T\nabla f_{\tau,\tau'}(x_i, x_{ij'}^+, x_{ik'}^-; \omega^*)$ defines a neural tangent kernel (NTK) in the network parameter space. Our bilevel formulation also indicates that hard triplet samples are essential for defining support vectors and outliers in OC-SVMs within NTK spaces, with their degree of difficulty measured using Lagrangian multipliers $\{\alpha_{ijk}\}$ as sample weights.

**Alternating Optimization.** To solve Equation (6), we simply learn $\{\alpha_{ijk}^*\}$ and $\omega^*$ as follows:

Step 1: Randomly sample triplets from the training dataset;
Step 2: Compute the solution $\{\alpha_{ijk}^*\}$ of the dual form of the OC-SVM in the LL problem;
Step 3: Update $\omega$ using SGD as the UL solution $\omega^*$ based on the solution $\{\alpha_{ijk}^*\}$;
Step 4: Repeat Step 1-3 until the UL objective converges.

### 3.3 ANALYSIS

**Lemma 1** (Contrastive Learning as NTK Regression). *Suppose that contrastive learning updates the model parameter $\omega$ as $\omega_{t+1} = \omega_t - \eta_t\nabla\ell(\omega_t) = \omega_t - \eta_t\sum_{i,j,k}\alpha_{ijk}^{(t)}\nabla f_{\tau,\tau'}(x_i, x_{ij}^+, x_{ik}^-; \omega_t)$ to minimize some contrastive loss $\ell(\omega)$, where $\alpha_{ijk}^{(t)} \ge 0$ denotes the sample weight for each training triplet $(x_i, x_{ij}^+, x_{ik}^-)$ at the $t$-th iteration and function $f$ is differentiable (everywhere). Assuming that the learning rates, $\{\eta_t\}$, satisfy $\lim_{t\to\infty}\eta_t = 0, \sum_{t=0}^{\infty}\eta_t = \infty, \sum_{t=0}^{\infty}\eta_t^2 < \infty$, then given a test triplet $(\tilde{x}, \tilde{x}^+, \tilde{x}^-)$, it holds that at the $T$-th iteration,*

$$f_{\tau,\tau'}(\tilde{x}, \tilde{x}^+, \tilde{x}^-; \omega_T) \le A - \sum_{t=0}^{T-1}\eta_t\left[\sum_{i,j,k}\alpha_{ijk}^{(t)}\kappa_{\omega_t}((x_i, x_{ij}^+, x_{ik}^-), (\tilde{x}, \tilde{x}^+, \tilde{x}^-))\right], \tag{7}$$

*where $A = \sup\left(f_{\tau,\tau'}(\tilde{x}, \tilde{x}^+, \tilde{x}^-; \omega_0) + O\left(\sum_{t=0}^{T-1}\eta_t^2\right)\right)$, provided that $f_{\tau,\tau'}(\tilde{x}, \tilde{x}^+, \tilde{x}^-; \omega_0)$ for any triplet $(\tilde{x}, \tilde{x}^+, \tilde{x}^-)$ is bounded.*

*Proof.* Based on local linear approximation and the assumptions in the lemma, we have

$$f_{\tau,\tau'}(\tilde{x}, \tilde{x}^+, \tilde{x}^-; \omega_{t+1}) - f_{\tau,\tau'}(\tilde{x}, \tilde{x}^+, \tilde{x}^-; \omega_t) = O(\eta_t^2) - \sum_{i,j,k}\alpha_{ijk}^{(t)}\kappa_{\omega_t}((x_i, x_{ij}^+, x_{ik}^-), (\tilde{x}, \tilde{x}^+, \tilde{x}^-)).$$

Now by summing up over $t$ from 0 to $T - 1$ recursively, we can complete our proof. $\square$

In practice, a loss function with a neural network as $f$ can be taken as a differentiable function and $\eta_t = O(\frac{1}{t})$ can easily satisfy the assumption. This lemma also indicates that contrastive learning can be viewed as an approximation of an OC-SVM with multiple NTKs in the network parameter space.

**Relation to Max-Margin Contrastive Learning.** To make sure that the distance from the positive sample, $d(x^+, x; \omega)$, is as small as possible compared with that from a negative sample, $d(x^-, x; \omega)$, we need to minimize $f_{\tau,\tau'}(\tilde{x}, \tilde{x}^+, \tilde{x}^-; \omega_T)$. Based on Lemma 1, we have a direct result as follows:

$$\min f_{\tau,\tau'}(\tilde{x}, \tilde{x}^+, \tilde{x}^-; \omega_T) \equiv \sum_{t=0}^{T-1} \eta_t \left[ \max \left\{ \sum_{i,j,k} \alpha_{ijk}^{(t)} \kappa_{\omega_t} \left( (x_i, x_{ij}^+, x_{ik}^-), (\tilde{x}, \tilde{x}^+, \tilde{x}^-) \right) \right\} \right], \quad (8)$$

where the RHS can be viewed as a maximum margin, learned within multiple NTK spaces at each iteration where each anchor $x_i$ introduce a kernel. That is, minimizing the distance between a positive pair and a negative pair is equivalent to maximizing a (weighted) margin with multiple NTKs.

**Different** from the literature of max-margin contrastive learning, such as (Shah et al., 2022), we aim to understand the behavior of contrastive learning from a geometric view of local linear approximations of the loss landscape, and accordingly learn to optimize gradients in backpropagation. To the best of our knowledge, we are the *first* to conduct such a study, leading us to different:

- *Reproducing Kernel Hilbert Space (RKHS):* Due to the gradient, our RKHS is the network parameter space, while a much smaller network output space is used in (Shah et al., 2022).
- *Kernel Methods:* We introduce OC-SVMs to learn optimal gradients with no labels, while (Shah et al., 2022) uses binary SVMs to select hard negative samples.
- *Theorems:* Our theorem reveals a strong connection between contrastive learning and (max-margin) kernel methods with multiple NTKs, which is missing in the current literature.

# 4 SINCE: SPARSE INFONCE LOSS FOR EFFICIENT SOLUTIONS

Similar to (Shah et al., 2022), tackling our bilevel optimization problem directly in deep learning proves to be highly challenging in practice. The vast RKHS, with its millions of dimensions, poses significant computational and storage difficulties on hardware like GPUs. To mitigate this issue, we introduce a novel contrastive loss, SINCE, designed to approximate the solutions of our GOAL.

**Motivation.** In fact, since the LL problem in Equation (6) is a convex problem, we can use projected gradient descent (PGD) to compute the dual solution, $\boldsymbol{\alpha}^* = \{\alpha_{ijk}^*\}$, as follows:

$$\boldsymbol{\alpha}_{t'+1} = \text{Proj}_\Delta \left( \boldsymbol{\alpha}_{t'} - \lambda_{t'} \left( \mathbf{K}_{\omega^*}(x_i) \boldsymbol{\alpha}_{t'} - \mathbf{f}_t(x_i) \right) \right) = \text{Proj}_\Delta \left( \lambda_{t'} \mathbf{f}_t(x_i) + (\mathbf{I} - \lambda_{t'} \mathbf{K}_{\omega^*}(x_i)) \boldsymbol{\alpha}_{t'} \right),$$
$$(9)$$

where at the $t'$ iteration, $\mathbf{K}_{\omega^*}(x_i) = [\kappa_{\omega^*}(\mathcal{X}_{ijk}, \mathcal{X}_{ij'k'})]$ stands for the NTK matrix for the anchor $x_i, \forall i$, $\mathbf{f}_t(x_i) = [f_{\tau,\tau'}(x_i, x_{ij}^+, x_{ik}^-; \omega_t)]$ for a vector, $\mathbf{I}$ for an identity matrix, $\lambda_{t'} \geq 0$ for a proper learning rate, and $\text{Proj}_\Delta$ for the projection-onto-simplex operator that can be conducted efficiently, *e.g.,* Chen & Ye (2011). However, in our case with very high dimensional RKHS, it is not practical to use many iterations to compute $\boldsymbol{\alpha}^*$. To address these issues, based on Chen & Ye (2011) we alternatively use the one-step approximation of Equation (9) with $\boldsymbol{\alpha}_0 = \mathbf{0}$ as shown below:

$$\boldsymbol{\alpha}^* \approx \boldsymbol{\alpha}_1 = \text{Proj}_\Delta \left( \lambda_0 \mathbf{f}_t(x_i) \right) = \max \left\{ \mathbf{0}, \lambda_0 \mathbf{f}_t(x_i) - \mu_t \mathbf{1} \right\}, \quad (10)$$

where $\mu_t$ is a scalar that is determined by the vector $\lambda_0 \mathbf{f}_t(x_i)$ and $\mathbf{1}$ is a vector of ones. In summary, *the solution of the OC-SVM can be approximated based on entry-wise rescaling followed by thresholding.*

**Loss Formulation: InfoNCE with Thresholding.** Based on our analysis above, we propose a strategy of *thresholding first and then normalization* for InfoNCE to approximate the OC-SVM solutions. This is equivalent to preserving "harder" triplets with larger $f$ values and removing "easier" ones, leading to a binary mask for each $\mathbf{f}_t(x_i)$. Accordingly, we formally define our SINCE loss as

$$\ell_{SINCE} = \mathbb{E}_x \left[ \log \sum_{(x^+, x^-)} \exp \left\{ f_{\tau,\tau'}(x, x^+, x^-; \omega) \right\} \cdot 1_{\{f_{\tau,\tau'}(x, x^+, x^-; \omega) \geq \mu_x\}} \right], \quad (11)$$

Table 2: Test accuracy comparison with the linear probe protocol.

| | CIFAR-10 | | | | | | STL-10 | | | | | |
|---|---|---|---|---|---|---|---|---|---|---|---|---|
| | # triplets | | | | | | # triplets | | | | | |
| | 20 | 40 | 60 | 80 | 100 | 16,256 | 20 | 40 | 60 | 80 | 100 | 16,256 |
| InfoNCE | 28.56 | 28.99 | 23.46 | 36.91 | 36.92 | 57.75 | 27.48 | 28.93 | 35.58 | 33.70 | 35.09 | 50.65 |
| **GOAL** | **30.22** | **34.62** | **38.79** | **45.13** | **49.41** | - | **31.91** | **42.70** | **45.38** | **44.17** | **46.66** | - |
| **SINCE** | 30.28 | 36.37 | 25.42 | 38.14 | 41.29 | **58.84** | 28.87 | 30.02 | 37.67 | 36.67 | 37.73 | **52.65** |

where $\mu_x$ is a predefined threshold, and $1_{\{\cdot\}}$ is an indicator function returning 1 if the condition holds, otherwise, 0. Note that instead of using $\mu_x$ in our experiments, which has an indeterminate range of values beforehand, we introduce another predefined parameter, $\gamma \in [0, 1]$, to control the ratio of triplets to be removed. This approach allows us to efficiently construct binary masks in Equation (11).

## 5 EXPERIMENTS

### 5.1 IMAGE CLASSIFICATION

We follow the representation learning and linear probe protocol (Oord et al., 2018; He et al., 2016; Yeh et al., 2021) for image classification to conduct comprehensive experiments on CIFAR-10 (Krizhevsky et al., 2009), STL-10 (Coates et al., 2011), and ImageNet-100 (Chun-Hsiao Yeh, 2022) datasets.

Table 3: Performance improvements (%) using SINCE over InfoNCE, with all triplets.

| | CIFAR-10 | STL-10 | ImageNet-100 |
|---|---|---|---|
| SimCLR | 1.09 | 2.00 | 2.46 |
| MOCO | 2.54 | 4.19 | 2.24 |
| BYOL | 2.69 | 3.36 | 2.53 |

**Datasets.** We take the labeled part for self-supervised pretraining without label leaking. We create a toy dataset CIFAR-10-toy by sampling 25% data from the original dataset for pretraining to mitigate the training overload, while for STL-10 we utilize its training data with no change. The downstream linear evaluation is made on the original test data in both CIFAR-10 and STL-10. We randomly sample an ImageNet-100 dataset from the ImageNet-1K dataset (Deng et al., 2009).

**Baselines.** We employ SimCLR (Chen et al., 2020a), MOCO (He et al., 2020), and BYOL (Grill et al., 2020) with ResNet-18 (He et al., 2016) as the backbone encoder for CIFAR-10 and STL-10, but with ResNet-50 for ImageNet-100. We compare our approach with InfoNCE loss to demonstrate its effectiveness of SINCE.

**Training Protocols.** In our GOAL and SINCE, we utilize Euclidean distances in Equation (1). We train our approach and baseline methods for 50 epochs with batch size 64, SGD optimizer with a momentum of 0.9, and weight decay of $10^{-4}$. we conduct our experiments on an Intel(R) Xeon(R) Silver 4214 CPU@2.20GHz and a single Nvidia Quadro RTX 6000 with 24GB

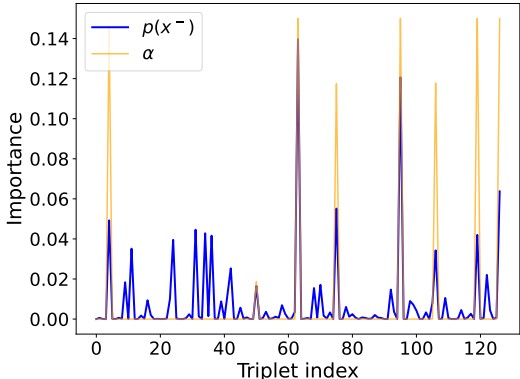

Figure 2: Comparison on gradient feature weights from InfoNCE as $p(x^-)$, and our GOAL as $\alpha$.

memory. We apply CVXOPT (Vandenberghe, 2010) to solve the LL problem in Equation (6) for GOAL, which runs on the CPU. We implement our algorithm and baseline methods based on the work of (Peng et al., 2022). Following the small-scale benchmark (Chen et al., 2020a; Yeh et al., 2021; Peng et al., 2022), we set both temperatures $\tau, \tau'$ to 0.07. We use a cosine-annealed learning rate of 0.5 for InfoNCE. The hyperparameter $C$ in Equation (6) is set to 0.15 for CIFAR-10 and 0.17 for STL-10 with slightly fine-tuning. For SINCE, we set $\gamma = 0.1$ in all the experiments.

**Evaluation Protocols.** Following the same setting as in (Peng et al., 2022) we train a linear classifier for each method. Specifically, after self-supervised pretraining, we freeze the network except for the last fully connected layer. We train the last-layer classifier in a supervised way using the full dataset. The linear classifier is trained for 50 epochs with a learning rate of 10.0, a batch size of 512, and a momentum of 0.9 in SGD for all experiments. We report the best performance of each method.

**Results.** We summarize our results from three aspects as follows:

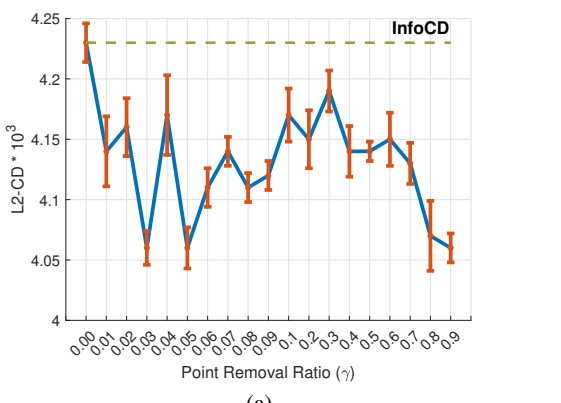 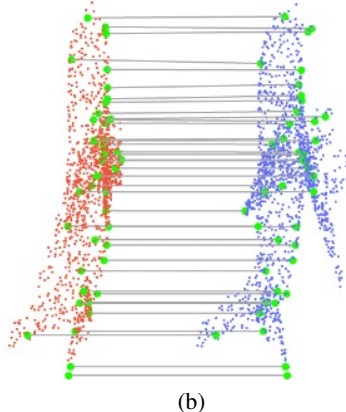

(a)                                        (b)

Figure 3: On ShapeNet-Part using CP-Net: **(a)** L2-CD *vs.* point removal ratio (smaller is better); **(b)** An illustration of matched point pairs preserved with $\gamma = 0.9$ for an airplance point cloud.

- *Sample Weight Comparison:* We illustrate a comparison of sample weights for gradients in InfoNCE and our GOAL for the same 127 triplets with the same $x, x^+$ in Figure 2. The feature extraction network is pretrained with 60 samples in each mini-batch on STL-10. As we see, the extremely high values of $p(x^-)$ and $\alpha$ co-occur quite frequently. For instance, the peak values around the 63rd triplet are 0.14 and 0.15 in InfoNCE and GOAL, respectively. Such observations are widely made when comparing the weights from both approaches. Therefore, the co-occurrences of large values in $p(x^-)$ and $\alpha$ indicate that the triplets that decide the boundaries of SVMs are almost those that contribute most to the gradient update in contrastive learning. In other words, we observe that *InfoNCE can produce good estimators for the solutions of OC-SVMs in SGD iterations.*
- *InfoNCE* vs. *GOAL* vs. *SINCE:* Table 2 lists our comparison results on CIFAR-10 and STL-10, where "-" indicates no results using all triples due to the hardware limit and running time. Although a smaller number of triplets would reduce the top-1 accuracy in the linear probe, our GOAL can significantly outperform both InfoNCE and SINCE in such cases. Using only 100 triplets per iteration, our GOAL can achieve performance that is close to both InfoNCE and SINCE with the full set of triplets. Besides, the performance of GOAL seems to be boosted more significantly than the other two with increasing number of triplets, which may benefit more for few-shot learning.
- *InfoNCE* vs. *SINCE for Self-Supervised Learning:* Table 3 shows the performance improvements achieved by our SINCE method with various network backbones for self-supervised learning on several benchmark datasets. In our experiments, we did not observe a significant difference in running time between the methods, as the number of images was relatively small.

### 5.2 3D POINT CLOUD COMPLETION

We demonstrate the effectiveness and efficiency of our SINCE loss by comparing with the recently proposed InfoCD Lin et al. (2023), which achieves the state-of-the-art for point cloud completion. To apply Equation (11) to the formulation of InfoCD, without loss of generality, letting $y_{ik}, y_{ik'}$ be two points in the ground-truth point cloud and $x_i = [x_{ij}]$ be the completed point cloud returned by some network with parameters $\omega$, we can define $f$ in Equation (11) as follows:

$$f_{\tau,\tau'}(x_i, y_{ik}, y_{ik'}; \omega) = \frac{1}{\tau'} \min_j \|x_{ij} - y_{ik}\| - \frac{1}{\tau} \min_j \|x_{ij} - y_{ik'}\|. \tag{12}$$

That is, for each ground-truth point, we search for the nearest neighbor in the point cloud returned by the completion network, and use the distance difference of an arbitrary pair as function $f$.

**Datasets & Backbone Networks.** We conduct our experiments on the **five** benchmark datasets: PCN (Yuan et al., 2018), MVP (Pan et al., 2021), ShapeNet-55/34 (Yu et al., 2021), ShapeNet-Part (Yi et al., 2016), and KITTI (Geiger et al., 2012). We compare our method using **thirteen** different existing backbone networks: FoldingNet (Yang et al., 2018), PMP-Net (Wen et al., 2021), PoinTr (Yu et al., 2021), SnowflakeNet (Xiang et al., 2021), CP-Net (Lin et al., 2022), PointAttN (Wang et al., 2022), SeedFormer (Zhou et al., 2022), PCN (Yuan et al., 2018), PFNet (Huang et al., 2020), TopNet

Table 5: Results on LiDAR scans from KITTI dataset under the Fidelity and MMD metrics.

| | FoldingNet | HyperCD+F. | InfoCD+F. | SINCE+F. | PoinTr | HyperCD+P. | InfoCD+P. | SINCE CD+P. |
|---|---|---|---|---|---|---|---|---|
| Fidelity ↓ | 7.467 | 2.214 | 1.944 | **1.887** | 0.000 | 0.000 | 0.000 | 0.000 |
| MMD ↓ | 0.537 | 0.386 | 0.333 | **0.305** | 0.526 | 0.507 | 0.502 | **0.453** |

(Tchapmi et al., 2019), MSN (Liu et al., 2020), Cascaded (Wang et al., 2020), and VRC (Pan et al., 2021), where we replace the CD loss with our SINCE wherever it occurs.

**Training & Evaluation Protocols.** We modify the public code[1] by replacing the InfoCD loss with our SINCE loss. For fair comparison, we strictly follow the experimental settings in InfoCD (Lin et al., 2023), including the same hyperparameters such as learning rate and its scheduler, regularization parameter, number of epochs, random seed, and batch size and order. We run all the comparisons on a server with 10 NVIDIA RTX 2080Ti 11G GPUs. Following the literature, we evaluate the best performance of all the methods using vanilla CD (lower is better). We also use F1-Score@1% (higher is better) to evaluate the performance on ShapeNet-55/34. For KITTI, we utilize the metrics of Fidelity and Maximum Mean Discrepancy (MMD) for each method (lower is better for both metrics).

**Results.** We first show our performance comparison on the ShapeNet-Part (Yi et al., 2016) dataset using CP-Net Lin et al. (2022) as the backbone network. We illustrate our results in Figure 3. As we see in (a), it is clear that thresholding can significantly improve the performance of InfoCD that is equivalent to our SINCE with $\gamma = 0$, in all the tested cases. In (b), we visualize the top 10% pairs of matched points between a completed point cloud (left) and its ground truth (right) in terms of Euclidean distance, which. These points can already capture well the global structures of the point clouds, which may lead to a better regularizer in training. Here, we set $\gamma = 0.9$ in all point cloud experiments without further tuning.

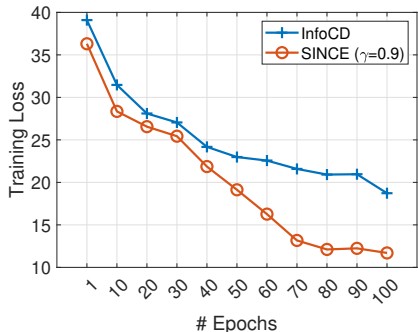

Figure 4: Training loss comparison on ShapeNet-Part using CP-Net.

Figure 4 illustrates the training loss curves of InfoCD and our SINCE with $\gamma = 0.9$, where we have normalized the binary masks for both for fair comparison. As we see, SINCE converges significantly faster than InfoCD with much lower losses, leading to better performance. As for running time, InfoCD takes $454.0 \pm 7.5$ seconds per epoch, while SINCE takes $480.0 \pm 4.9$ seconds per epoch.

We also summarize detailed comparison results in Table 4, Table 5, Table 6, Table 7, and Table 8, where SINCE outperforms InfoCD in all the cases, leading to new state-of-the-art results. Note that for KITTI, we follow (Xie et al., 2020) to finetune the models on ShapeNetCars (Yuan et al., 2018) and evaluate them on KITTI.

Table 4: Average per-point L1-CD×1000 on PCN.

| Networks | InfoCD | SINCE |
|---|---|---|
| FoldingNet | 12.14 | **11.31** |
| PMP-Net | 7.92 | **7.87** |
| PoinTr | 7.24 | **7.21** |
| SnowflakeNet | 6.86 | **6.82** |
| PointAttN | 6.65 | **6.62** |
| SeedFormer | 6.52 | **6.46** |

## 6 CONCLUSION

In this paper, we aim to interpret deep contrastive learning from a geometric perspective by optimizing gradients in backpropagation. By drawing connections with OC-SVMs, we propose a new gradient-optimized contrastive learning (GOAL) approach based on bilevel optimization. In this approach, optimal gradients are learned through OC-SVMs as the lower-level problem, while the upper-level problem updates the network weights using SGD based on these optimal gradients. We also reveal a strong connection between contrastive learning and kernel methods with multiple NTKs. Furthermore, we introduce a new SINCE loss to address the computational challenges of GOAL for large-scale learning. We demonstrate the superior performance of our approach in the tasks of image classification and point cloud completion.

**Limitations.** Thresholding in SINCE may introduce additional computational burdens in learning, and GOAL has not yet reached its full potential in real-world applications such as few-shot learning. We will investigate both aspects in future work.

---

[1]https://github.com/Zhang-VISLab/NeurIPS2023-InfoCD

Table 6: Completion results on MVP in terms of L2-CD $\times 10^4$ ($\downarrow$) and EMD $\times 10^2$ ($\downarrow$).

| | Methods | airplane | cabinet | car | chair | lamp | sofa | tabel | watercraft | bed | bench | bookshelf | bus | guitar | motorbike | pistol | skateboard | Avg. |
|---|---|---|---|---|---|---|---|---|---|---|---|---|---|---|---|---|---|---|
| CD | PCN | 4.50 | 8.83 | 6.41 | 13.01 | 21.33 | 9.90 | 12.86 | 9.46 | 20.00 | 10.26 | 14.63 | 4.94 | 1.73 | 6.17 | 5.84 | 5.76 | 9.78 |
| | InfoCD+PCN | 3.95 | 8.82 | 6.38 | 12.03 | 17.43 | 9.63 | 12.41 | 8.69 | 18.92 | 8.75 | 13.40 | 5.02 | 1.84 | 6.06 | 5.81 | 4.37 | 9.41 |
| | **SINCE+PCN** | **3.76** | **8.65** | **6.19** | **11.84** | **17.24** | **9.45** | **12.22** | **8.52** | **18.73** | **8.56** | **13.22** | **4.84** | **1.67** | **5.87** | **5.68** | **4.15** | **9.23** |
| | TopNet | 4.12 | 9.84 | 7.44 | 13.26 | 18.64 | 10.77 | 12.95 | 8.98 | 19.99 | 9.21 | 16.06 | 5.47 | 2.36 | 7.06 | 7.04 | 4.68 | 10.30 |
| | InfoCD+TopNet | 3.98 | 9.81 | 7.42 | 13.24 | 17.87 | 10.52 | 12.45 | 8.93 | 19.69 | 8.52 | 14.62 | 5.42 | 2.35 | 7.05 | 6.52 | 4.21 | 10.01 |
| | **SINCE+TopNet** | **3.74** | **9.57** | **7.18** | **13.02** | **17.61** | **10.27** | **12.23** | **8.68** | **19.44** | **8.32** | **14.39** | **5.18** | **2.14** | **6.86** | **6.34** | **3.99** | **9.78** |
| | MSN | 2.73 | 8.92 | 6.50 | 10.75 | 13.37 | 9.26 | 10.17 | 7.70 | 17.27 | 6.64 | 12.10 | 5.21 | 1.37 | 4.59 | 4.62 | 3.38 | 7.99 |
| | InfoCD+MSN | 7.28 | 8.51 | 6.03 | 10.18 | 12.91 | 8.87 | 9.72 | 7.24 | 16.82 | 6.21 | 11.67 | 4.79 | 0.91 | 4.15 | 4.17 | 2.97 | 7.56 |
| | **SINCE+MSN** | **6.98** | **8.24** | **5.78** | **9.92** | **12.60** | **8.55** | **9.40** | **7.01** | **16.43** | **5.92** | **11.14** | **4.21** | **0.81** | **3.86** | **3.88** | **2.68** | **7.28** |
| | Cascaded | 2.54 | 8.62 | 5.93 | 8.76 | 11.22 | 8.46 | 9.20 | 6.61 | 14.63 | 6.09 | 10.17 | 4.95 | 1.55 | 4.34 | 4.23 | 3.19 | 7.25 |
| | InfoCD+Cascaded | 2.43 | 8.05 | 5.73 | 8.77 | 10.47 | 8.24 | 9.18 | 6.41 | 14.37 | 6.02 | 10.45 | 4.70 | 1.45 | 4.23 | 4.16 | 2.99 | 7.12 |
| | **SINCE+Cascaded** | **2.32** | **7.94** | **5.62** | **8.64** | **10.35** | **8.16** | **9.07** | **6.28** | **14.25** | **5.90** | **10.42** | **4.58** | **1.32** | **4.10** | **4.04** | **2.87** | **7.01** |
| | VRC | 2.20 | 7.92 | 5.60 | 7.49 | 8.15 | 7.45 | 7.52 | 5.20 | 11.90 | 4.88 | 7.39 | 4.53 | 1.15 | 3.90 | 3.44 | 3.22 | 6.09 |
| | InfoCD+VRC | 2.03 | 7.88 | 5.41 | 7.31 | 7.92 | 7.22 | 7.30 | 5.01 | 11.67 | 4.65 | 7.14 | 4.30 | 0.97 | 4.68 | 3.19 | 3.04 | 5.87 |
| | **SINCE+VRC** | **1.94** | **7.43** | **5.15** | **7.03** | **7.62** | **7.01** | **7.03** | **4.75** | **11.41** | **4.34** | **6.87** | **4.02** | **0.91** | **4.41** | **2.96** | **2.78** | **5.62** |
| EMD | PCN | 4.70 | 7.99 | 5.75 | 6.90 | 11.99 | 5.32 | 6.60 | 5.40 | 9.84 | 4.85 | 7.87 | 5.24 | 10.56 | 4.93 | 4.86 | 5.59 | 6.80 |
| | InfoCD+PCN | 3.75 | 5.59 | 3.97 | 5.23 | 10.11 | 4.42 | 5.45 | 4.47 | 7.29 | 4.21 | 5.55 | 3.53 | 6.12 | 4.02 | 4.70 | 3.84 | 5.17 |
| | **SINCE+PCN** | **3.22** | **5.03** | **3.43** | **4.72** | **9.54** | **3.88** | **4.91** | **4.12** | **6.75** | **3.65** | **5.00** | **3.02** | **5.57** | **4.39** | **4.16** | **3.29** | **4.63** |
| | TopNet | 4.89 | 6.30 | 4.07 | 7.01 | 10.75 | 6.47 | 7.50 | 4.68 | 8.09 | 6.27 | 6.80 | 3.50 | 4.21 | 4.26 | 6.02 | 3.49 | 6.18 |
| | InfoCD+TopNet | 4.47 | 6.02 | 3.81 | 6.82 | 10.21 | 6.05 | 7.12 | 4.37 | 7.87 | 5.87 | 6.02 | 3.31 | 4.06 | 4.11 | 5.82 | 3.15 | 5.72 |
| | **SINCE+TopNet** | **4.02** | **5.66** | **3.43** | **6.44** | **9.82** | **5.67** | **6.76** | **4.01** | **7.51** | **5.48** | **5.65** | **2.95** | **3.68** | **4.74** | **5.45** | **2.77** | **5.35** |
| | MSN | 2.75 | 4.02 | 3.47 | 4.44 | 6.28 | 3.74 | 4.46 | 3.82 | 5.27 | 3.34 | 4.28 | 2.92 | 2.07 | 3.30 | 3.62 | 2.21 | 3.94 |
| | InfoCD+MSN | 2.18 | 3.51 | 2.97 | 3.96 | 5.77 | 3.21 | 3.92 | 3.24 | 4.75 | 2.86 | 3.79 | 2.41 | 1.50 | 2.81 | 3.09 | 2.64 | 3.38 |
| | **SINCE+MSN** | **1.95** | **3.28** | **2.73** | **3.72** | **5.53** | **3.02** | **3.68** | **3.02** | **4.51** | **2.62** | **3.54** | **2.18** | **1.27** | **2.57** | **2.85** | **2.41** | **3.15** |
| | Cascaded | 3.03 | 6.82 | 5.44 | 5.16 | 7.55 | 5.57 | 4.73 | 4.88 | 6.85 | 3.51 | 5.71 | 5.81 | 5.30 | 4.30 | 4.42 | 3.44 | 5.18 |
| | InfoCD+Cascaded | 2.87 | 6.23 | 5.39 | 5.06 | 7.10 | 5.45 | 4.57 | 4.79 | 6.42 | 3.49 | 5.15 | 5.72 | 3.58 | 4.19 | 4.27 | 2.91 | 5.01 |
| | **SINCE+Cascaded** | **2.52** | **6.05** | **5.17** | **5.01** | **7.02** | **5.32** | **4.41** | **4.63** | **6.21** | **3.31** | **5.02** | **5.47** | **3.42** | **4.10** | **4.11** | **2.75** | **4.85** |
| | VRC | 3.03 | 7.57 | 6.14 | 5.49 | 6.15 | 5.80 | 4.65 | 4.97 | 6.58 | 3.45 | 5.28 | 6.59 | 3.08 | 4.45 | 4.56 | 3.20 | 5.27 |
| | InfoCD+VRC | 2.68 | 7.26 | 5.83 | 5.15 | 5.82 | 5.49 | 4.36 | 4.68 | 6.22 | 3.13 | 4.97 | 6.26 | 2.77 | 4.13 | 4.15 | 2.89 | 4.97 |
| | **SINCE+VRC** | **2.47** | **7.07** | **5.64** | **4.95** | **5.63** | **5.30** | **4.17** | **4.47** | **5.96** | **3.02** | **4.76** | **6.05** | **2.55** | **3.91** | **4.01** | **2.78** | **4.78** |

Table 7: Results on ShapeNet-34 using L2-CD$\times 1000$ ($\downarrow$) and F1 score ($\uparrow$).

| Methods | 34 seen categories | | | | | 21 unseen categories | | | | |
|---|---|---|---|---|---|---|---|---|---|---|
| | CD-S | CD-M | CD-H | Avg. | F1 | CD-S | CD-M | CD-H | Avg. | F1 |
| FoldingNet | 1.86 | 1.81 | 3.38 | 2.35 | 0.139 | 2.76 | 2.74 | 5.36 | 3.62 | 0.095 |
| InfoCD + FoldingNet | 1.54 | 1.60 | 3.10 | 2.08 | 0.177 | 2.42 | 2.49 | 5.01 | 3.31 | 0.157 |
| **SINCE + FoldingNet** | **1.47** | **1.54** | **3.02** | **2.01** | **0.183** | **2.36** | **2.43** | **4.99** | **3.26** | **0.160** |
| PoinTr | 0.76 | 1.05 | 1.88 | 1.23 | 0.421 | 1.04 | 1.67 | 3.44 | 2.05 | 0.384 |
| InfoCD + PoinTr | 0.47 | 0.69 | 1.35 | 0.84 | 0.529 | 0.61 | 1.06 | 2.55 | 1.41 | 0.493 |
| **SINCE + PoinTr** | **0.41** | **0.65** | **1.28** | **0.78** | **0.534** | **0.61** | **1.02** | **2.51** | **1.37** | **0.496** |
| SeedFormer | 0.48 | 0.70 | 1.30 | 0.83 | 0.452 | 0.61 | 1.08 | 2.37 | 1.35 | 0.402 |
| InfoCD + SeedFormer | 0.43 | 0.63 | 1.21 | 0.75 | 0.581 | 0.54 | 1.01 | 2.18 | 1.24 | 0.449 |
| **SINCE + SeedFormer** | **0.41** | **0.62** | **1.20** | **0.74** | **0.583** | **0.52** | **1.02** | **2.12** | **1.21** | **0.452** |

Table 8: Results on ShapeNet-55 using L2-CD$\times 1000$ ($\downarrow$) and F1 score ($\uparrow$).

| Methods | Table | Chair | Plane | Car | Sofa | CD-S | CD-M | CD-H | Avg. | F1 |
|---|---|---|---|---|---|---|---|---|---|---|
| FoldingNet | 2.53 | 2.81 | 1.43 | 1.98 | 2.48 | 2.67 | 2.66 | 4.05 | 3.12 | 0.082 |
| InfoCD + FoldingNet | 2.14 | 2.37 | 1.03 | 1.55 | 2.04 | 2.17 | 2.50 | 3.46 | 2.71 | 0.137 |
| **SINCE + FoldingNet** | **2.06** | **2.28** | **1.01** | **1.43** | **2.02** | **2.14** | **2.45** | **3.38** | **2.65** | **0.141** |
| PoinTr | 0.81 | 0.95 | 0.44 | 0.91 | 0.79 | 0.58 | 0.88 | 1.79 | 1.09 | 0.464 |
| InfoCD + PoinTr | 0.69 | 0.83 | 0.33 | 0.80 | 0.67 | 0.47 | 0.73 | 1.50 | 0.90 | 0.524 |
| **SINCE + PoinTr** | **0.62** | **0.78** | **0.32** | **0.74** | **0.62** | **0.40** | **0.67** | **1.43** | **0.83** | **0.529** |
| SeedFormer | 0.72 | 0.81 | 0.40 | 0.89 | 0.71 | 0.50 | 0.77 | 1.49 | 0.92 | 0.472 |
| InfoCD + SeedFormer | 0.65 | 0.72 | 0.31 | 0.81 | 0.62 | 0.43 | 0.71 | 1.38 | 0.84 | 0.490 |
| **SINCE + SeedFormer** | **0.62** | **0.71** | **0.30** | **0.75** | **0.63** | **0.42** | **0.68** | **1.36** | **0.82** | **0.493** |

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
