# OpenReview forum: "GRADIENT-OPTIMIZED CONTRASTIVE LEARNING"
_ICLR.cc/2025/Conference — Submitted to ICLR 2025_

### Official Review · Reviewer_HErX · 2024-10-28

**Soundness:** 3
**Presentation:** 3
**Contribution:** 2
**Rating:** 6
**Confidence:** 3

**Summary:**

The paper is motivated by the $(\phi, \varphi)$ formulation of contrastive losses from Tian (2022), consider the gradient weighted by a linear combination of triplet gradient features, and address issues for easy triplets weighting down more important triplets. The paper connects this with OC-SVM to determine the gradient weights. The paper also introduce a more computational efficient approximation which applies hard triplet sampling. Report performance improvements over contrastive baselines on image-classification and 3d point cloud completion.

**Strengths:**

1. The paper is comprehensive and well-structured.
2. Addresses relevant issue in contrastive learning which is the weighting of triplets/pairs in the gradient.
3. The paper report good performance results over the contrastive InfoCD (Lin et.al, 2023) on 3d cloud completion.

**Weaknesses:**

1. The image classification results are too low compared to Peng et., (2022), Chen et.al., (2020a), Yeh et.al, (2021), which the authors base implementations details on, and to Max-Margin Contrastive Learning (Shah et al., 2022). It would more relevant if you could use the original datasets, train to convergence, and show improvement over the mentioned work with similar settings. The final results on ImageNet100 are also not included in Table 3.
2. GOAL presents computational difficulties and can rarely be realized fully in practice. The approximation SINCE develop into a hard negative sampling strategy (thresholding before normalizing) which, I think, is not more than a simple modification to the InfoNCE loss.

**Questions:**

See first point weaknesses.

---

> ### Author Response · Authors · 2024-11-27
>
> We thank the reviewer for valuable comments. Below are our responses to your concerns:
>
> **1. Lower performance for image classification:** As mentioned in lines 346-351, due to the computational complexity of GOAL, we only use a small portion of the training data to train our model to avoid excessive training time. Note that all experiments are conducted under the same settings, ensuring fair comparability of results. Table 3 highlights the improvements of SINCE over InfoNCE to save space. The detailed numbers are listed below:
>
> |                          | CIFAR-10 | STL-10 | ImageNet-100 |
>
> |----------------------------------------------------------------|
>
> | SimCLR | 57.75 | 50.65 | 26.14 |
>
> | SimCLR + SINCE | 58.84 | 52.65 | 28.60 |
>
> |----------------------------------------------------------------|
>
> | MOCO | 16.73 | 37.12 | 26.72 |
>
> | MOCO + SINCE | 19.27 | 41.31 | 28.96 |
>
> |---------------------------------------------------------------- |
>
> | BYOL | 43.96 | 49.85 | 37.76 |
>
> | BYOL + SINCE | 46.65 | 53.21 | 40.29 |
>
> |----------------------------------------------------------------|
>
> **2. SINCE is no more than a simple modification to the InfoNCE:** We respectfully disagree, as there are at least three key points to highlight:
>
> * *(a) Theoretical results:* The design of SINCE is fundamentally rooted in the need to approximate OC-SVM solutions effectively and efficiently, providing a deeper understanding of the method.
>
> * *(b) Empirical results:* SINCE consistently outperforms InfoNCE in our experiments. To the best of our knowledge, SINCE has not been proposed in the literature, and we welcome any references to existing work that is identical to ours.
>
> * *(c) Generalization:* The design principles of SINCE can be easily applied to other loss functions, such as those listed in Table 1, to enhance baseline performance.
>
> Together, these points strongly support the novelty of SINCE.

---

### Official Review · Reviewer_3QkF · 2024-10-31

**Soundness:** 3
**Presentation:** 3
**Contribution:** 3
**Rating:** 6
**Confidence:** 4

**Summary:**

This paper focuses on contrastive learning, where the authors propose a gradient-optimized contrastive learning framework, which enhances network training by optimizing gradient updates during backpropagation as a bilevel optimization problem. To address the
computational complexity of GOAL, they also develop propose a Sparse InfoNCE (SINCE) learning framework, which improves the Lagrangian multiplier approximation by incorporating hard triplet sampling into InfoNCE.

**Strengths:**

1. The proposed learning framework which enhances network training by optimizing gradient updates during backpropagation as a bilevel optimization problem provides some interesting insights.
2. The mathematical formulation and the theoretical grounding provided are thorough and robust. It advances the understanding of the mechanics behind gradient updates and their optimization in neural networks.

**Weaknesses:**

1. The learning framework proposed in the experimental section does not seem to have significant gains compared to existing frameworks. In addition, the experimental dataset is relatively small and the network structure is single (ResNet), so it is recommended to further supplement with larger benchmark sets and experiments with different network structures.
2. While the paper addresses computational efficiency, the actual computational cost, in terms of time and resources, required for implementing these models in a real-world setting is not thoroughly explored. For large-scale deployment, the computational overhead could still be a limiting factor, which may not be fully mitigated by the proposed Sparse InfoNCE loss.
3. While the paper compares the proposed method with some existing approaches, a broader comparison with other state-of-the-art methods could have been included to position the paper within the current research landscape more clearly.

**Questions:**

Besides the Weaknesses above, I also have the following questions:
1. In this paper, the authors use the UL problem to update network weights, while the LL problem to learn optimal gradients for SGD.  My question is, can the two be reversed? Why?
2. In the proposed Sparse InforaNCE, a predefined threshold is required. How does it choose? Is there a challenge in its choice?Suggestions for supplementing relevant sensitivity analysis experiments.
3. Is there anything missing from Table 3? I did not capture any information about SINCE.
4. In Section 3.3, the authors mentioned using OC-SVM to optimize gradients. My question is: what impact will different kernel functions have on performance? Suggest supplementing relevant experiments.

---

> ### Author Response · Authors · 2024-11-27
>
> We thank the reviewer for the valuable comments. Below are our responses to your concerns:
>
> **1.1. Modest improvement:** We would like to highlight that our goal in image classification is not to achieve state-of-the-art performance on each dataset. Instead, we aim to demonstrate (1) the computational issues in GOAL, and (2) the generalization and improvements of SINCE over InfoNCE. For point cloud completion, these results are considered state-of-the-art (SOTA) performance on benchmark datasets, where even small improvements are significant and challenging. For example, SeedFormer achieved a score of 6.74 on the PCN dataset, reducing the previous SOTA of 7.21 by SnowflakeNet by 0.47. More importantly, SINCE consistently enhances baseline performance.
>
> **1.2. More network backbones and larger datasets:**  For different backbone evaluations, we have assessed **13** different networks on **5** different datasets, including **KITTI** (a large-scale dataset with images and lidar scans) in the task of point cloud completion, as highlighted in lines 427-431. This evidence may alleviate your concerns.
>
> **2. Running time and overhead:** As stated in line 462-464, "InfoCD takes 454.0$\pm$7.5 seconds per epoch, while SINCE takes 480.0$\pm$4.9 seconds per epoch" on ShapeNet-Part using CP-Net, leading to about 6% overhead. Note that our implementation is unoptimized, and the overhead could be smaller with more careful coding.
>
> **3. State-of-the-art (SOTA) comparisons:** Tables 6, 7, and 8 present SOTA comparisons for point cloud completion, featuring **8** different networks evaluated on **3** benchmark datasets.
>
> **4. UL and LL reversion:** No, they cannot. As mentioned in lines 191-194, the upper-level (UL) problem is influenced by the **optimal** parameters (e.g., $\alpha$ values for OC-SVM) derived from the lower-level (LL) problem, which in turn is affected by the **non-optimal** parameters (e.g., network weights) from the UL problem.
>
> **5. Threshold in SINCE:** As stated in lines 333-336, we do not set the threshold directly due to the difficulty in adaptively determining the range of values based on data. Instead, we use a percentage to set the threshold and introduce a new parameter $\gamma$ in our experiments. The sensitivity analysis is shown in Fig. 3(a).
>
> **6. Missing results in Table 2:** As noted in lines 404-405, "-" indicates the absence of results using all triples due to hardware limitations and running time constraints. In other words, we are unable to run GOAL under this setting.
>
> **7. Kernels in OC-SVM:** As stated in lines 239-240, the kernel function must be linear over the triplet gradient features. This requirement stems from the primal form of the OC-SVM in Eq. 5, allowing $\Delta\omega$ to be represented as a linear combination of triplet gradient features for updating network weights in SGD.

---

> > ### Comment · Reviewer_3QkF · 2024-12-03
> > **Reply to the rebuttal**
> >
> > Thanks for the authors' efforts during the rebuttal period, which addressed most of my concerns. After reading the comments of other reviewers and the author's rebuttal, I maintained my original positive score.

---

### Official Review · Reviewer_W9Bd · 2024-11-01

**Soundness:** 2
**Presentation:** 2
**Contribution:** 2
**Rating:** 5
**Confidence:** 3

**Summary:**

The paper titled "Gradient-Optimized Contrastive Learning" (GOAL) introduces a contrastive learning framework aimed at improving gradient during backpropagation. GOAL reinterprets contrastive learning as a one-class support vector machine (OC-SVM) problem in neural tangent kernel (NTK) space, focusing on hard triplet samples to enhance model robustness. To this end, the authors propose a contrastive loss, Sparse InfoNCE (SINCE), which integrates hard triplet sampling for efficient learning. Experiments in image classification and point cloud completion demonstrate that SINCE enhances contrastive learning by making gradient updates more efficient and effective, achieving competitive results across multiple datasets.

**Strengths:**

1. Strong Empirical Results:
GOAL and SINCE demonstrate superior performance on benchmark datasets in both image classification and 3D point cloud completion. The results indicate that the proposed method outperforms traditional InfoNCE-based approaches, particularly in settings requiring robust gradient optimization.

2. Clear Contributions to Contrastive Learning:
The paper provides a strong conceptual link between contrastive learning and sparse kernel machines. This insight offers a fresh view that could impact future research in unsupervised and semi-supervised representation learning by enabling more efficient contrastive frameworks.

3. Comprehensive Evaluation:
Extensive experiments across multiple benchmark datasets, including CIFAR-10, STL-10, and ShapeNet-Part, showcase the robustness and adaptability of the proposed approach across different types of data and tasks (classification and point cloud completion), underscoring its versatility.

**Weaknesses:**

1. Insufficient Link Between Motivation and Proposed Solution:
In the motivation subsection (L035-053), the authors present various perspectives on contrastive learning, emphasizing the importance of hard-negative samples. However, the subsequent question posed—"How should we optimize the gradients in contrastive learning effectively and efficiently?"—lacks a clear justification. For example, from the analysis of Tian et al (2022) , it is unclear what is the specific issue taht motivates gradient optimization within contrastive learning. The rationale for optimizing gradients and how this links to the concerns raised in the motivation are not sufficiently clarified.

Similarly, the proposed approach in the "Approach" subsection (L065-080) could benefit from further explanation of why certain strategies or assumptions are adopted. For instance, while the authors introduce a bi-level optimization framework, it would be helpful to understand why this framework is necessary and what limitations of current methods necessitate this choice. A clearer justification of the selected strategy would aid in understanding and appreciating the method. Overall, establishing a stronger connection between the problem raised in the motivation and how the proposed approach addresses it would improve clarity. Additionally, the methods section could more explicitly address why each methodological choice is made, rather than focusing solely on what the choices are.

2. Structural Suggestions for Improved Clarity:
In the methods section (L182-189), the authors begin discussing the core problem; however, addressing this at the beginning of the paper might set a more coherent foundation for readers. In L188-189, the question "What if \(\alpha_{x, x^{+}, x^{-}}\) does not have an explicit form of \(\omega\)?" would benefit from a discussion of the conditions under which this situation might arise and why it is significant.
From lines L190-194, introducing and explaining the bi-level optimization framework in general would improve clarity. Specifically, a brief explanation of the upper-level and lower-level problems could help readers unfamiliar with the framework to understand the significance of this choice.

3. Interpretation of Key Inferences:
In lines L240-243, the authors infer that "Our bilevel formulation also indicates that hard triplet samples ...". However, this inference is not readily apparent from the preceding discussion. It would be valuable if the authors could elaborate on the indicators leading to this conclusion and provide a more explicit description.

4. Clarity in the Learning Process Description:
The description of the algorithm in sections 3 and 4 would benefit from additional clarity, specifically regarding the overall learning process. A pseudo-code algorithm summarizing the training pipeline could help to practically illustrate the proposed method.

5. Efficiency Considerations:
In Table 2, including a discussion of the overall wall-clock time, FLOPs, or MACs for the proposed approach would provide insights into the efficiency-performance trade-offs. Additionally, it raises questions about inefficiency compared to the standard InfoNCE version, as bi-level optimization methods are generally known to be computationally intensive.

6. Emphasis on Why Over What:
In its current form, the paper predominantly addresses *what* is being done rather than *why*. Strengthening the paper’s focus on the core problem being solved, while clearly linking the proposed solution to this problem, would significantly enhance its readability and impact.

**Questions:**

1. Could you offer more context on why specific assumptions and strategies (e.g., bi-level optimization) were chosen, as well as any alternatives they considered?

2. Could the authors provide additional details on the efficiency of the proposed method? For example, a comparison of wall-clock time, FLOPs, or MACs relative to the standard InfoNCE implementation would offer insights into potential performance-efficiency trade-offs.

3. In the bilevel formulation, the authors conclude that hard triplet samples serve as support vectors and outliers. Could the authors elaborate on this inference, including which indicators are being referenced and how they lead to this conclusion?

---

> ### Author Response · Authors · 2024-11-27
>
> We thank the reviewer for valuable comments, and we will polish the paper for further clarification. Below are our responses to your concerns:
>
> **W1.1. Insufficient Link Between Motivation and Proposed Solution:** In lines 44-47, we clearly state our reasoning, *"This behavior raises concerns about the effectiveness and robustness of the gradients in contrastive learning because useful (hard) negative samples can be easily buried among many non-useful (easy) negative samples, leading to similar weights for generating gradients."*
>
> **W1.2. & Q1. Why bilevel optimization approach?** In lines 182-195, we clearly state our motivation for proposing a bilevel optimization formula: *"Motivation: Sample weights for gradients and network weights may be fully coupled."* Bilevel optimization is a fundamental approach to address such problems. As an alternative, we propose SINCE as an effective and efficient approximation of the solutions for bilevel optimization.
>
> **W2. Structural Suggestions for Improved Clarity:** We will do so.
>
> **W3 & Q3. Interpretation of Key Inferences:** This indication comes from the definition of support vectors in SVMs, which we believe is a basic concept in machine learning. These support vectors define the "hard" triplets in our approach, as their Lagrange multipliers $\alpha$ are nonzero. We recommend the reviewer to refer to a machine learning textbook for more details.
>
> **W4. Clarity in the Learning Process Description:"** Please refer to lines 245-249.
>
> **W5 & Q2. Efficiency Considerations:** As stated in line 462-464, "InfoCD takes 454.0$\pm$7.5 seconds per epoch, while SINCE takes 480.0$\pm$4.9 seconds per epoch" on ShapeNet-Part using CP-Net. The inefficiency of bilevel optimization led us to propose SINCE as an alternative.
>
> **W6. Emphasis on Why Over What:** We respectfully disagree, as our submission addresses all of your "why" questions directly. In summary, our submission addresses both the "what" and the "why" questions as thoroughly as possible. We will strive to make this clearer in our final version to improve readability, as you suggested.

---

### Official Review · Reviewer_XXkx · 2024-11-03

**Soundness:** 3
**Presentation:** 3
**Contribution:** 3
**Rating:** 6
**Confidence:** 3

**Summary:**

The paper makes two contributions. First, it introduces a novel approach to contrastive learning known as Gradient-Optimized Contrastive Learning (GOAL), which reformulates the contrastive learning problem as a bilevel optimization challenge. Additionally, it enhances the computational efficiency of GOAL through the implementation of a new loss function called Sparse InfoNCE. Theoretically, the paper establishes the validity of this approach by establishing a lemma that shows the equivalence between contrastive learning and One-class Support Vector Machine (OC-SVM) using multiple Neural Tangent Kernels (NTKs). Empirically, the method is evaluated on tasks such as image classification and 3D point-cloud completion.

**Strengths:**

Overall, the paper is well-written and presents a fresh perspective on the contrastive learning problem by formulating it as a bilevel optimization challenge. While the derivation of the Upper Level (UL) is relatively straightforward, the formulation of the Lower Level (LL) as an OC-SVM problem is intriguing. This approach not only highlights the relationship between hard triplet samples and outliers in OC-SVMs but also adds depth to the theoretical framework. Furthermore, the paper introduces an easy-to-implement algorithm that is tested across a wide range of scenarios, demonstrating its versatility and effectiveness.

**Weaknesses:**

Firstly, the experimental results on image classification are limited to just three small datasets and two essentially similar architectures from the same ResNet base. To enhance this section, it would be beneficial to include results from more modern architectures, such as EfficientNet [1] and PyramidNet [2], as well as from more challenging datasets such as CIFAR100 or Stanford Cars [3]. Additionally, while the experiments on point-cloud completion cover a diverse range of settings, the proposed method shows only a modest improvement over InfoCD. Besides, other concerns are addressed in the questions below.

**Questions:**

+ In lines 177-178, it is stated that "the gradient $\triangle \omega_t$ is computed by a linear combination of triplet features. While this holds true for the InfoNCE loss as derived in the previous section, is it also applicable to state-of-the-art contrastive learning methods? Clarification on this point would be valuable.

+ In lines 183-184, could you elaborate on the motivation for mentioning "when the calculation of each $\alpha^{(t)}_{(x, x^+, x^-)}$ relies on the triplet features"? According to Equation (3), it appears that $\alpha$ does not depend on the triplet features, and further explanation would help clarify this statement.

+ Could you provide additional context for using the dual regularized OC-SVM as the Lower Level (LL) objective to find the optimal $\alpha^*_{i, j, k}$? Including a few sentences about this motivation before Equation (5) would enhance the communication of the idea, along with a derivation of the dual form or a reference to the dual form of the classical OC-SVM problem, as presented in Equation (6).

+ It seems that there was a slight mathematical mismatching between Lemma 1 and Eq. (8). In particular, the minimum of the LHS should be equal to the maximum of the summation

$$\max \sum \eta_t [\sum \alpha^{(t)}_{ijk} \kappa (\cdot,\cdot)]$$

instead of the summation of the maximum

$$\sum \eta_t [\max (\sum \alpha^{(t)}_{ijk} \kappa (\cdot,\cdot))]$$

+ Could you clarify why the two results in Table 2 were omitted? Additionally, it appears that some entries in this table, such as the results for GOAL with 20 and 40 triplets, may have been mistakenly bolded.

+ Could you provide more detail on how the function $f$ was selected for the image classification task?

**References:**

[1] Tan, M., & Le, R. P. (2019). EfficientNet: Rethinking model scaling for convolutional neural networks. arXiv preprint arXiv:1905.11946.

[2] Han, D., Wang, J., & Yin, S. (2017). PyramidNet: Nested deep learning models for image classification. arXiv preprint arXiv:1610.02915.

[3] K. Krause, H. Stark, J. Deng, & L. Fei-Fei. (2013). 3D Object Representations for Fine-Grained Categorization. In *Proceedings of the IEEE Conference on Computer Vision and Pattern Recognition (CVPR)*. Retrieved from http://ai.stanford.edu/~jkrause/cars/car_dataset.html

---

> ### Author Response · Authors · 2024-11-27
>
> We thank the reviewer for the valuable comments. Below are our responses to your concerns:
>
> **1. Network backbones and datasets for image classification:** Please note that our goal in image classification is not to achieve state-of-the-art performance on each dataset. Instead, we aim to demonstrate (1) the computational issues in GOAL, and (2) the generalization and improvements of SINCE over InfoNCE. For different backbone evaluations, we have assessed **13** different networks on **5** different datasets, including **KITTI** in the task of point cloud completion, as highlighted in lines 427-431. This evidence may alleviate your concerns regarding image classification. For challenging image datasets, we believe that ImageNet-100 is more demanding than CIFAR-100 and Stanford Cars. More information about ImageNet-100 can be found at https://www.kaggle.com/datasets/ambityga/imagenet100. We will try to include more network backbones and datasets for image classification in the final version.
>
> **2. Modest improvement over InfoCD:** These results are considered state-of-the-art (SOTA) performance on benchmark datasets, where even small improvements are significant and challenging. For example, SeedFormer achieved a score of 6.74 on the PCN dataset, reducing the previous SOTA of 7.21 by SnowflakeNet by 0.47. More importantly, SINCE consistently enhances baseline performance.
>
> **3. Clarification on "the gradient $\Delta\omega$ is computed by a linear combination of triplet features":** This statement holds for the losses in Table 1, so called $(\phi, \psi)$-contrastive losses that are widely used in the literature. I am not sure which SOTA contrastive method you are referring to, but through the calculation of gradients, you should be able to verify it.
>
> **4. Motivation for mentioning "when the calculation of each $\alpha_{(x,x^+,x^-)}^{(t)} relies on the triplet features":** There are generally two ways to compute $\alpha$:
>
> * *(a) No dependency on triplet features, for example, InfoNCE in Eq. 3:* In such cases, as discussed in L45-53, the corresponding gradients may be dominated by irrelevant data, causing the useful information (i.e., the hard triplets that are crucial for updating networks) to be overlooked.
>
> * *(b) Dependent on triplet features:* As proposed in our submission, to identify these hard triplets for optimizing gradients, we consider learning $\alpha$ locally based on the triplet features, as illustrated in Figure 1. This approach leads to GOAL and SINCE.
>
> **5. OC-SVM:** The choice of OC-SVM is based on the format of triplet loss in Eq. 1, as there should be no difference in labeling for individual triplets $(x,x^+,x^-)$. OC-SVM aims to learn gradients to directly reduce the triplet loss over iterations in gradient descent. Scholkopf et al. (1999), as cited in the submission, provides comprehensive information about the primal and dual formulations of OC-SVM.
>
> **6. Mismatching between Lemma 1 and Eq. (8):** Note that all the $\eta$'s are nonnegative, and thus the two formulas are equivalent.
>
> **7. Missing results and mistakenly bolded in Table 2:** As stated in L404-405, "-" indicates no results using all triples due to the hardware limit and running time. In order words, we are not able to run GOAL under this setting. We will correct the bold font in the table.
>
> **8. Function $f$ for image classification:** We did not select particular function but just plugged the distance function in the original SimCLR, MOCO, or BYOL into $f$ with triplets.

---

### Official Review · Reviewer_gX9d · 2024-11-04

**Soundness:** 3
**Presentation:** 4
**Contribution:** 4
**Rating:** 6
**Confidence:** 4

**Summary:**

This paper proposes a novel method that modifies InfoNCE but provides better gradient during training with SGD. From the observation that sample weights and network weights are coupled together, they tried to optimize both objectives via bilevel optimization. They connect sample weights optimization with one-class support vector machine then use these weights to optimize network weights, which is novel approach. They also provide practical version of this approach, namely SINCE, which is equivalent to thresholding with predefined threshold. They showed empirical performance on image classification and 3D point cloud completion.

**Strengths:**

1. I think the approach they provide is novel. As they mentioned in the introduction, they inherit the core motivation of previous negative sampling works but tried to connect these concepts with one-class support vector machine.
2. The writing is clear and cohesive.
3. Empirical results are powerful.

**Weaknesses:**

1. I enjoyed reading upto section 3, however, the final form of SINCE seems too simple, arbitrary, and not tight. Now we need to find appropriate hyperparameters. According to Table2, SINCE performs way much poor than GOAL especially large triplet cases. Figure 3(a) it seems the performance is quite sensitive to these parameters

**Questions:**

1. Related to weakness: Is there a better way to better approximate GOAL, such as higher-order approximation? Since GOAL approach seems novel and powerful (from Table 2), I wish to witness better approximation showing better result than current SINCE. It doesn't need to be full experiments (Derivation with simple experiments would be fine). I would happy to raise my score if this concern is resolved.
2. I wish to check the triplets provided in Figure 2  - especially 7 peaks high in both cases and ~10 peaks that are only appeared with InfoNCE. Can we distinguish those samples in our eyes? In other words, are those separation corresponds to our inception?

---

> ### Author Response · Authors · 2024-11-26
>
> Thank you for the valuable comments. Below are our responses to your concerns:
>
> **1. A Better Approximation of GOAL:** In this paper, the proposed GOAL primarily focuses on theoretical analysis for contrastive learning. However, there are several practical issues in our implementation that hinder its widespread use: (1) Gradient feature extraction is time-consuming for large networks with millions or billions of parameters; (2) Kernel computation is also time-intensive; (3) Solving the lower-level (LL) constrained quadratic programming problem requires significant CPU time; (4) Storing the extracted features demands substantial GPU memory. Therefore, any approximation of GOAL should address these issues. Additionally, since the LL problem is already a quadratic programming (i.e., second-order) problem, higher-order approximations are not applicable in our case.
>
> In addition to the motivation from gradient descent discussed in Section 4, we can also consider the approximation from the primal formulation. Drawing inspiration from SVM primal solvers like Pegasos (Shalev-Shwartz et al., ICML 2007), our hard thresholding in SINCE is well-suited for online learning of neural networks using sparse gradient combinations. This offers another perspective on why SINCE outperforms InfoNCE.
>
> Lastly, we introduced a temperature parameter, $\tau$, into SINCE, similar to InfoNCE, and reran all the experiments for image classification. The results presented in our submission are based on this updated approach with $\tau=1$. Below are the results:
>
> |                          | CIFAR-10 | STL-10 | ImageNet-100 |
>
> |----------------------------------------------------------------|
>
> | SimCLR | 57.75 | 50.65 | 26.14 |
>
> | SimCLR + SINCE ($\tau$=1) | 58.84 | 52.65 | 28.60 |
>
> | SimCLR + SINCE ($\tau$=0.1) | 59.11 | 53.52 | 29.77 |
>
> |----------------------------------------------------------------|
>
> | MOCO | 16.73 | 37.12 | 26.72 |
>
> | MOCO + SINCE ($\tau$=1) | 19.27 | 41.31 | 28.96 |
>
> | MOCO + SINCE ($\tau$=0.1) | 20.53 | 42.54 | 30.89 |
>
> |---------------------------------------------------------------- |
>
> | BYOL | 43.96 | 49.85 | 37.76 |
>
> | BYOL + SINCE ($\tau$=1) | 46.65 | 53.21 | 40.29 |
>
> | BYOL + SINCE ($\tau$=0.1) | 47.79 | 55.06 | 41.61 |
>
> |----------------------------------------------------------------|
>
> Clearly, having this hyperparameter can constantly improve the performance of SINCE, which may be considered as a better approximation. We will update the numbers accordingly in the final version.
>
> **2. Are those separation corresponds to our inception?** This is a very interesting question. Unfortunately, we cannot find clear or strong connections between those separated triplets and human inception. This can be attributed to the fact that all the triplets are mapped into a very high-dimensional space, which may be entirely different from human inception.

---

### Meta-Review · Area_Chair_gTbt · 2024-12-20

**Metareview:**

This paper introduces a novel perspective on contrastive learning by formulating it as a bilevel optimization problem. The authors observe that the gradient of InfoNCE—the widely used loss in contrastive learning—is a weighted combination of triplet (positive/negative pair) gradients. Instead of relying on the explicit dependence of these weights on the form of the loss, the authors propose to optimize the weights through a bilevel formulation. To address the computational cost of solving the bilevel problem, they also develop an efficient approximation-based formulation.

Empirically, the proposed methods are compared against InfoNCE on several tasks, including image classification and point cloud completion. All reviewers acknowledged the merits of the work but raised concerns about the limited scope of the empirical evaluations, which do not fully explore the broader applicability of the proposed method.

The AC agrees with the reviewers that, despite its merits, the work falls short of providing a comprehensive evaluation, and hence sadly recommends `rejection`. On a related note, recent advances in bilevel optimization suggest that first-order methods can achieve faster convergence (see, for example, Liu et al., *BOME! Bilevel Optimization Made Easy: A Simple First-Order Approach*, NeurIPS'22).

We hope this review helps the authors identify areas for improvement, particularly in expanding the empirical evaluations and exploring state-of-the-art methods for bilevel optimization.

**Additional Comments On Reviewer Discussion:**

Reviewers raised several concerns, primarily regarding the extent and scope of the experiments. While the authors expanded their experiments during the author-reviewer discussion period, the additions did not adequately address large-scale problems where contrastive learning typically excels. As a result, none of the reviewers championed the paper, despite its merits. The AC agrees with the reviewers on this point and, therefore, recommends rejection.

---

### Decision · Program_Chairs · 2025-01-22

Reject